# Use of TLS (LiDAR) for Building Diagnostics with the Example of a Historic Building in Karlino

**Rafał Nowak [1],\*** , **Romuald Orłowicz [1]** and **Radosław Rutkowski [2]**

[1] Faculty of Civil Engineering and Architecture, West Pomeranian University of Technology, Szczecin, 70-311 Szczecin, Poland; orlowicz@yandex.ru

[2] Faculty of Economics and Transport Engineering, Maritime University of Szczecin, 70-500 Szczecin, Poland; szczecin.ar@gmail.com

\* Correspondence: rnowak@zut.edu.pl; Tel.: +48-605-642-800

**Abstract:** This article presents the use of TLS (LiDAR) measurement for the evaluation of the technical conditions of a historic building. A FARO M70 laser scanner was used in the study. The measurements was taken as an RCP point cloud. The measurement allowed to partially determine the cause of the building damage. The performed measurement allows to propose a precise solution that could be pre-fabricated. The study shows the usefulness of TLS in building diagnostics. Improper measurement could lead to a wrong solution and a certain degree of uncertainty.

**Keywords:** LIDAR; TLS; 3D scan; case study; measure; management; quality

## 1. Introduction

The diagnostics of historical buildings is an important issue. Often, the conditions of these buildings raise numerous concerns. These objects, which are usually under the care of the monuments' conservator, require special attention due to their historical relevance. The majority of analyses of existing structures are based on visual inspection, searching for visible cracks on the walls or slabs and others [1–4]. In some cases, special equipment is used, including sclerometers, ultrasonic emitter, radar or other non-destructing material testing methods [5–8]. Less popular are high-precision measuring techniques used only in case of uneven settling or extreme cases (excessive deflections of slabs). If necessary, core samples are taken from walls or structural parts for precise determination of concrete strength [9–14]. Furthermore, 3D FEM models can be applied for the prediction of the causes of the existing state (possible cracking). The models help to prepare proper strengthening solution considering all historical building requirements [15–22]. In many cases, even though cracks occur, the construction still has capacity reserves, making repairs unnecessary. It is particularly important to repair structures only when it is necessary because each repair changes the historical value of the structure [23]. This is important for the preservation of historical heritage.

The process of obtaining geometric data is a key step in the evaluation, design and implementation of work related to the general improvement of the technical condition of existing buildings. The acquisition of geometric data are based on the studies of engineering geodesy. The basic characteristic of a building is its size. Therefore, issues related to the acquisition of geometric data of buildings are often called large-size metrology. Intensive development of methods and measuring systems for large-size structures took place in the 1960s. Methods were initially developed for large-sized steel structures (mainly off-shore).

These methods and systems were then successively transferred to civil engineering. Initially, they were based on optical measuring devices (levels, theodolites). Measurement methods dedicated to large-size objects, such as the reference line method and optical levelling, are some of the basic methods

used for measurement [24]. Measurement with the use of optical devices and the aforementioned methods are still in use today. Along with the development of measurement technologies, measurement methods based on advanced electronic measuring devices have been introduced. The most common type of electronic total stations implements the polar measurement.

The introduction of total stations has revolutionized the process of obtaining geometric data of large-sized objects. The process of improving the measuring method is still ongoing. Currently, the intensely developed 3D scanning technology (the so-called Light Detection and Ranging—LiDAR, TLS—Terrestrial Laser Scanning) provides enormous possibilities. Currently, the use of laser technology in buildings diagnostics and geometry measuring is gaining interest but is still not as popular due to equipment costs. Several studies have applied TLS scanning in material diagnostics [25–28]. TLS scanning is also used for analysing deflections in construction [29–32].

TLS measuring has some limitations, including weather conditions [33], external temperature, vibrations and the ability of materials to reflect. It is impossible to measure reflective surfaces or glass objects (also windows). Even reflective veneer on furniture can create issues for the equipment. All mirror or glass object after registration have to be manually cut out, sometimes windows. Similar issues occur for any type of laser measuring devices such as total stations or rangefinders. TLS scans from stationary scanners are made from singular positions, which means that the data from all measuring points must be complied together after the tests. There are several techniques supporting those data analyses. One of the best techniques is to use the "target method"; however, it requires many changes of the targets position (sometimes it is the only possible method). The more convenient method is using "point to cloud" registration, which searches between scans for similarities. This type of registration requires low distances between measuring points (about 5–10m) and does not allow to create an analysis for continuous surfaces like walls without visible corners. It also demands large-scale overlaps of points between scans.

A good process of registration depends a lot on the quality of the scanner but also the experience of the operator. Scanner quality settings like point cloud density and ranging noise quality can hugely influence the final results. The final result of the TLS scans is dependent on proper compilation (connecting scans together). The results can then be compared with measurements performed with other equipment or reference (survey) points. Even with high-end equipment with high resolution, the overall registration quality can be hindered if the registration process has not been performed properly.

The basic disadvantage of TLS is the cost of its application. The cost of purchasing a scanner with a measurement uncertainty of less than 1 mm as well as appropriate equipment, including computer and software, to enable efficient handling of point clouds is very high. Another problem is the fact that a small number of available programs are still able to cope with point cloud analysis, and an even smaller number do it in a dynamic way. Currently, probably only Autodesk's Revit can dynamically load point clouds. In practice, the possibility of opening point clouds is often limited by the RAM of the computer. Thus, all 3D scans over 16 GB for basic computers create significant problems during analysis. Considering that in the case of 3D scans of larger objects, files of the order of 200 GB are often encountered, this problem becomes prevalent. The advantages of the process of building diagnostics and the process of designing and performing repair described in this article outnumber the potential disadvantages. This is a technology that will certainly dominate diagnostic and repair works of buildings in the near future.

In the case described in this paper, the TLS technology was used to obtain the geometric data [29–31,34,35], which allowed to conduct a detailed and effective diagnostics of the analysed building. It was part of diagnostic methods chosen for execution of Expert Opinion for an analysed building. The study presents successful TLS data acquisition to diagnose the cause of degradation of walls of a building. The window lintel line was used as the reference line for building the current deformation state. Additionally, the gable wall verticality and the floor positions in cross section of building were also considered. The deflection of floor beams was also checked. The provided results were confirmed with on-site investigations.

## 2. Case Description

The object of research was a building with a simple cubical shape, with three storeys and a partial basement (Figure 1). The second and third storeys are occupied by residential dwellings. The ground floor is currently not in use; previously, it was used as a public utility, e.g., a police station. The owner of the building (Karlino Commune) plans to rebuild the ground floor and adapt it as a kindergarten.

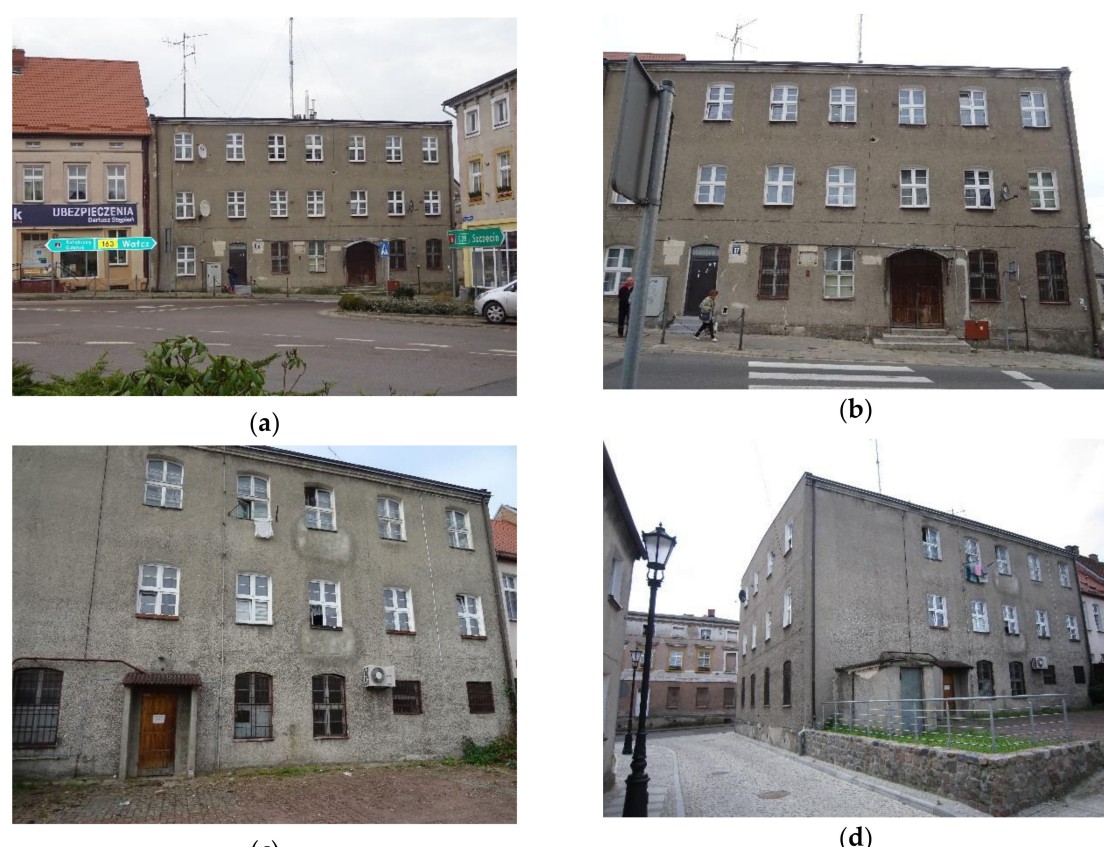

**Figure 1.** View of the building: (**a**) front; (**b**) front; (**c**) back; (**d**) side.

The studied building has been rebuilt several times. The history of the building dates back to the beginning of the 19th century. Only the cellars and foundations have survived to this day. The current area of the building was originally occupied by two tenement houses. Figure 2a shows the condition of the buildings from the period after World War II, with its original division of the space. Figure 2b shows the reconstructed design, which has been preserved to this day. The reconstruction of the building was carried out on the old foundations of both houses. Both foundations of the previously existing tenement houses were on two different levels. The accepted solution was wrong. The difference in the foundation levels in both parts of the building can cause uneven settlement. A better solution would be a complete demolition, including foundation and a whole-scale reconstruction. Alternatively, one could consider improving the foundation of a smaller tenement house (connecting building).

The initial inspection of the building showed only a single vertical crack on the facade (Figure 3). The cause of the crack was, at that time, unknown. Further investigation of the archival photographs (Figure 2) allowed to determine the reasons. The crack was located at the point where the two original tenement houses were connected. The crack in the building was visible from both the courtyard and the front elevation. It can be assumed that the building spontaneously split itself at the location of foundations difference. Apart from the crack, no other significant signs of structural damages were visible on the outside. The structure of the building was not designed to withstand such dilatation.

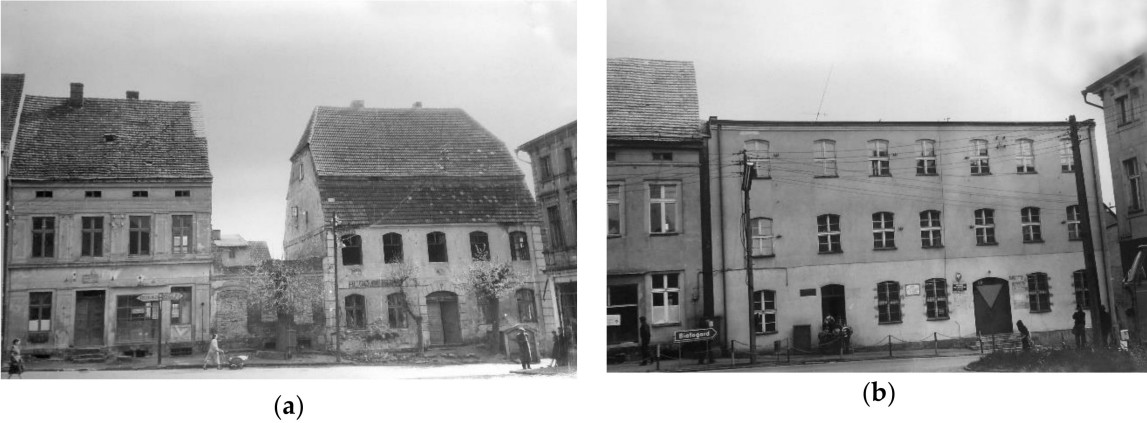

**Figure 2.** Archival photos: (**a**) before the reconstruction; (**b**) after the reconstruction.

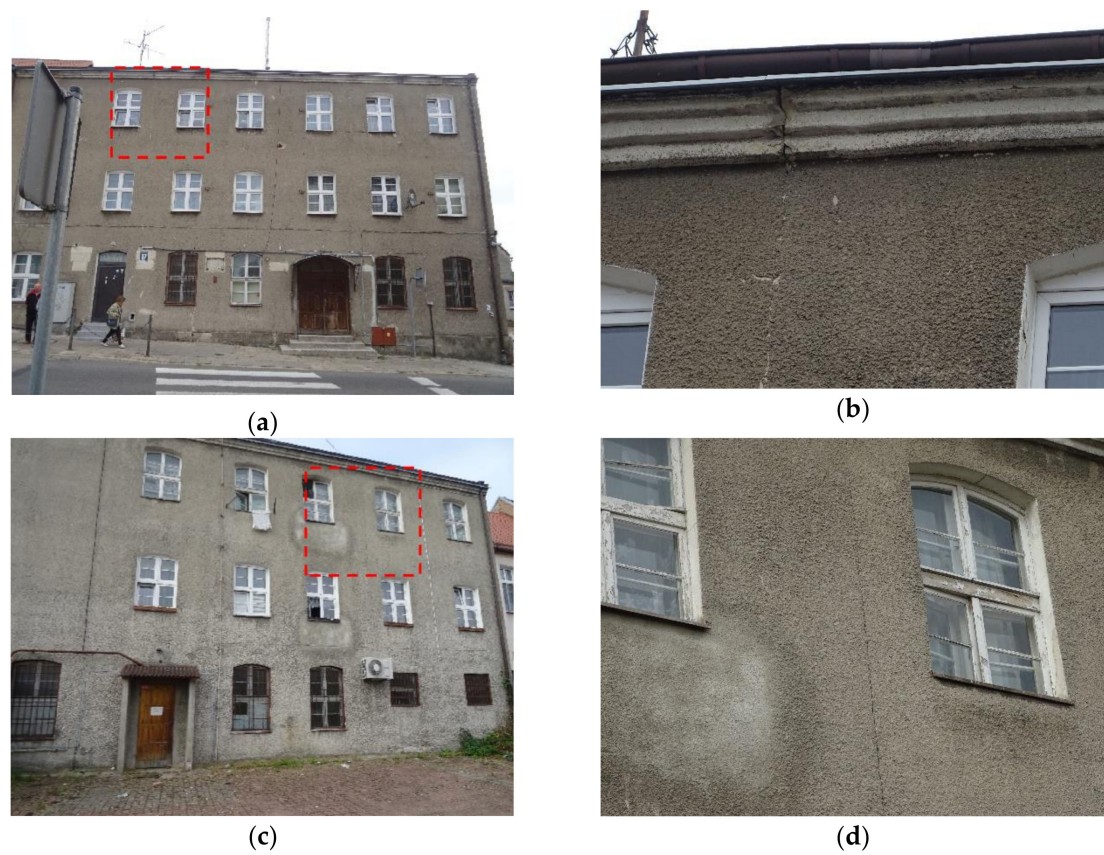

**Figure 3.** Damaged facade: (**a**) front; (**b**) zoom on crack in the front; (**c**) back; (**d**) zoom on the crack in the back.

Further visible damage to the structure of the building was identified during internal observations. Numerous plaster cracks were observed on walls and ceilings. Particularly important were significant vertical cracks in the walls and cracks under the ceiling (Figure 4).

The visible cracks on the plaster required further examination. The plaster was removed, which allowed to determine the state of the construction layer. The action confirmed structural damages. The bricks of a load-bearing walls were damaged—they had been split by the shear force (Figure 5a). The following step focused on the evaluation of the ceiling between the ground and the first floor. The ceiling was identified as a beam and block floor (Figure 5b)—a type based on steel beams with reinforced concrete slabs. Unfortunately, this type of floor has a relatively low load capacity and has rarely been used in residential buildings.

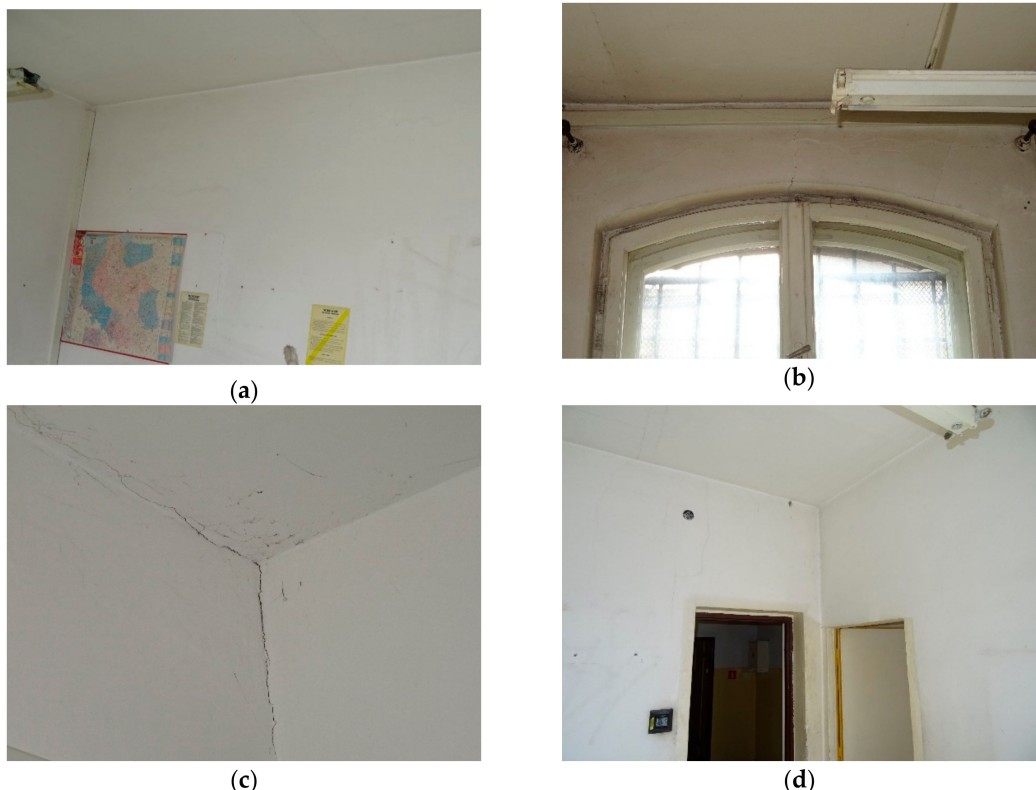

**Figure 4.** Interior damages: (**a**) wall cracks; (**b**) damages near the masonry arch lintel; (**c**) wall corner damages; (**d**) cracks near the ceiling base.

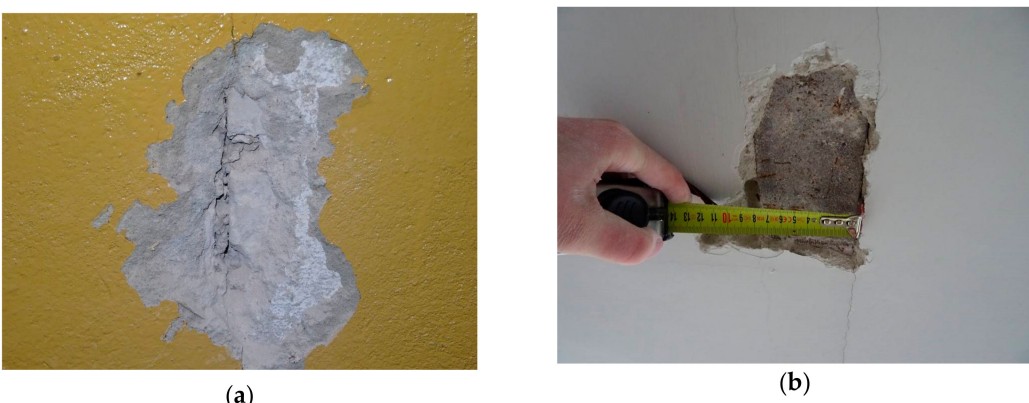

**Figure 5.** Uncovered main structure elements: (**a**) vertical crack of the brick wall; (**b**) measurement of uncovered steel beam.

The beam support on load-bearing walls was also verified. The inspection showed that there was no tie beams in the building; the beams were located directly on the walls. This situation explains the problem of cracks in the walls under the floor beams. It was found that the shear resistance of the wall was exceeded and the steel beams were supported incorrectly. An, intermediate plate or reinforced concrete tie beams should have been installed. Incorrect fixing of the floor beams caused vertical cracks on the walls, which began at the floor beams support location. The lack of tie beams is disadvantageous to the entire load-bearing structure as no elements are binding the walls together. The result is a lowered stability of the whole building. In older buildings, the role of tie beams was often played by steel anchors fixed to the wooden beams and supporting walls. The inspection did not confirm existence of any type of strengthening.

As part of the evaluation, an analysis of the internal staircase was performed. The staircase was located in the part of the building that originally coincided with a smaller tenement house. Diagnostics at this stage were performed using a HILTI PS 200 S (Figure 6) inductive scanner. The evaluation determined the reinforcement as φ10 with a 10 cm spacing and partition rods of φ6.

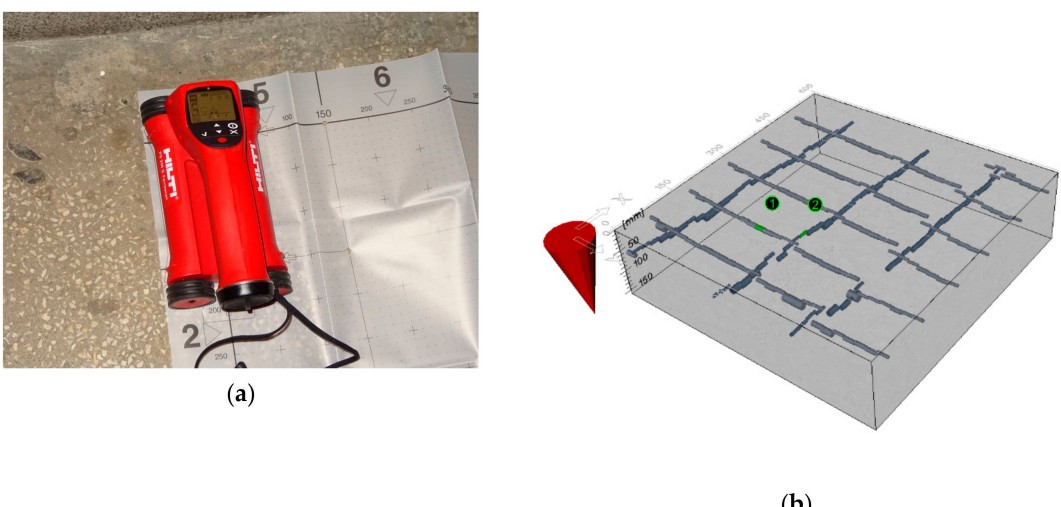

(**a**)

(**b**)

**Figure 6.** Staircase measurement: (**a**) during measurement; (**b**) view of digital model made in HILTI PROFIS Detection.

## 3. Geometric Measurements

Obtaining geometric data is a key element of building diagnostics. Precise measurements are particularly useful for the evaluation of a historic building, which change unpredictably with the flow of time. The data is necessary for designing and choosing proper strengthening methods.

Electronic total stations and polar measurement methods are commonly used to carry out these necessary evaluations. The methods allows to obtain the coordinates of any visible point of the building in the assumed coordinate system (example [36]). In order to obtain a complete geometry extensive number of measurements is required. Each point of measurement requires a set up of the equipment and potential calibration. The model of the building is acquired after aligning network of points from all measurements, including the external and internal ones. The calculations are time-consuming and have a certain measurement bias. The errors occurring during measurements at individual stations and during the implementation of the points matrix follow the propagation of the uncertainty law.

High precision of measurements is necessary for proper evaluation of the building conditions and further selection of repair methods. The considered case study is an example of this, as the repair solution required installing a steel reinforcement structure. The correct design of the geometry of the reinforcing structure and its execution and assembly tolerances depends on the available geometric data. The influence of measurement accuracy on the designed tolerances is shown in Figure 7. As can be seen, increasing the measurement uncertainty increases the so-called uncertainty band. This reduces the possible design and assembly errors, almost denying the possibility of making the structure on the site.

A promising alternative to total station measurements is the increasingly popular TLS technology. Scanners with a high resolution allow to obtain close to complete geometry of the measured object. The amount of geometric data of the building is incomparably greater than with the use of total stations. Creating virtual models of objects taking into account the external and internal geometry is easier. The method allows to perform full and detailed diagnostics.

TLS technology was used to evaluate the studied building. At first, it was impossible to properly assess the cause of the damage. The diagnosis of the condition could not be performed precisely due

to several matters. It was impossible on the basis of traditional observations and engineering practices. It was not until the use of the 3D scanning that the real cause of building damage was uncovered.

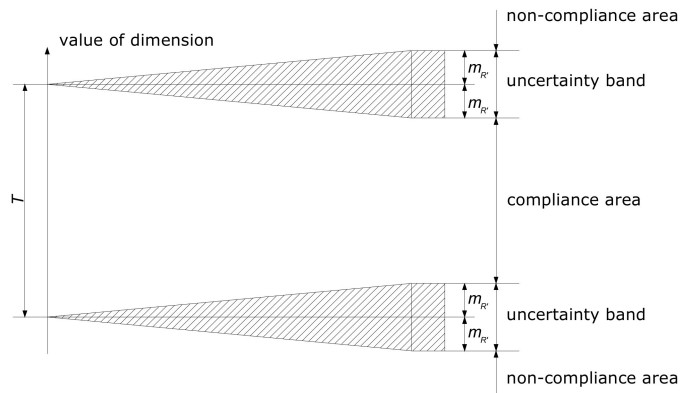

**Figure 7.** The impact of measurement inaccuracy on the appraisal of measured element dimensional quality. mR'—standard deviation occurring in the determination of dimension R, T—process tolerance, compliance area—tolerance area less the standard deviation occurring in the determination of dimension R, non-compliance area—area outside the tolerance area plus the standard deviation of measurements, uncertainty band—area where compliance cannot be determined.

A 3D stationary scanner with a 360° field of view and a measurement uncertainty <0.2 mm Focus M70 by FARO was used. Uncertainty was determined individually for the scanner used by the manufacturer on a testing ground with basic parameters—the distance to the target was 22–23 meters, and the distance between the standard measuring points, 5 meters. The obtained accuracy is fully sufficient for the assumed needs.

Originally, the diagnosis of the building was attempted to be carried with a total station using the polar method. However, the obtained measurement results did not give a full picture of the current geometry of the whole object. Attempts to establish a measuring matrix inside the building to perform the measurements and combine them with the results of external dimensions measurements aroused great reservations. The initial analysis of errors of measurement work in the scope of transfer of errors in setting up subsequent measurement bases showed that the final accuracy of the measurement work is practically of the same order as the predicted deformations of the examined object. Such a state could not be accepted. The decision to use other measurement methods was natural. TLS measurements were practically the only alternative. The conducted TLS measurements and analysis of errors of these measurements showed a definite advantage of this method over total station measurements for the analysed case. The specified errors of the TLS measurement process are based on manufacturer's data and parameters obtained from the software of the scanner. The approach utilizes the "point to cloud" function available in the manufacturer's software. After the measurement compilation, the application produces calculations for measurement and compilation errors for each scan individually and the final result. The measurements errors did not exceed 3 mm in the study, which, in the opinion of the authors, is an acceptable value with regards to the total building displacement.

## 4. TLS Measurements and Diagnosis

The measurements were made for the whole external body of the building and the geometry of the internal floors with basements (Figures 8 and 9). As a result of the measurements, complete three-dimensional geometrical data of the object were obtained, which allowed to carry out a detailed analysis of its technical condition. External measurements were made in the full-colour spectrum while interior was scanned in monochrome. Colour measurement requires additional measurement with photographs in order to superimpose colour textures on the resulting point clouds, which extends the measurement process.

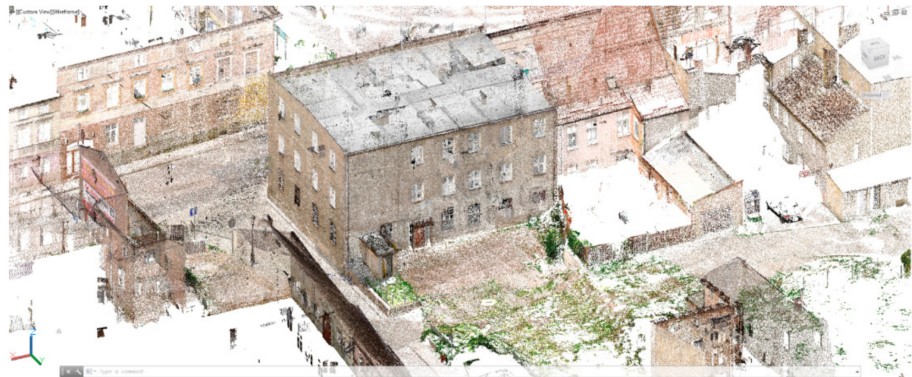

**Figure 8.** Outside view of the building made using TLS in Autocad 2015—axonometry.

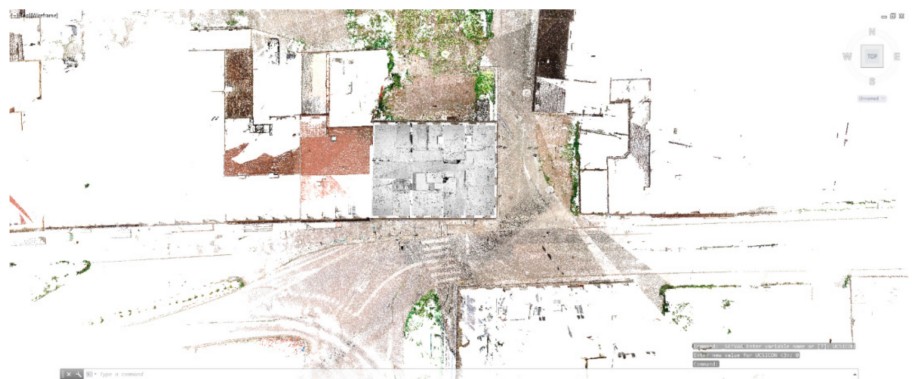

**Figure 9.** Outside view of the building made using TLS in Autocad 2015—aerial view.

Full diagnostics of the building geometry allowed to identify additional—invisible for standard methods—aspects of the studied building.

The measurements of the front wall showed the building to be tilted to the right by approx. 5 cm (Figure 10). The result was similar for the lintel window lines of the ground floor and the second floor. For the first floor, the value was lower and probably resulted from the initial construction errors. During the analysis of the building geometry, it was also found that the front wall deflected from the vertical plane by about 2 cm and the cornice of the building was heavily distorted. Similar results were obtained from both sides of the building. The largest identified deformations were found in the front wall. No significant deformations of the building were observed on the end wall. Taking into account the results obtained, a conclusion was drawn that the building evenly leaned towards the street (south–eastern direction).

Analysis of the geometry was performed in Autocad 2015 software based on the Robin Cloud Platform (RCP). Levelling was performed automatically in FARO SCENE at the stage of assembling point clouds from single scans.

In addition to the measurements, tenants were interviewed to establish the most recent building history. On the basis of the obtained information, it was found that several years earlier, construction work was carried out in the immediate vicinity of the building. During the work, the building was excavated significantly. The period of construction work coincides with the occurrence of cracks in the walls on the top floor. The investor was asked to gather the documentation and the construction log. After verifying the ordinates of the construction work around the building, it was found that the excavation could have been carried out below the ordinate of the building foundations (Figure 11). The ordinate of the building foundations is located at about +14.00 m above sea level and the work were performed at +12.87 m above sea level, according to Figure 11. The work was performed to replace the sewage system and modernize the road. The deformations of the building identified suggest that this work was performed incorrectly, without proper protection of the walls or conducting works in parts. It is also possible that the ground was incorrectly compacted after the construction

work was carried out. Polish law does not require supervision of an engineer with a construction licence, even though ground work can have a direct impact on neighbouring buildings. The work was not supervised. Additionally, it was found that the building on the adjacent side of the crossroads in the immediate vicinity of the work was also visibly damaged.

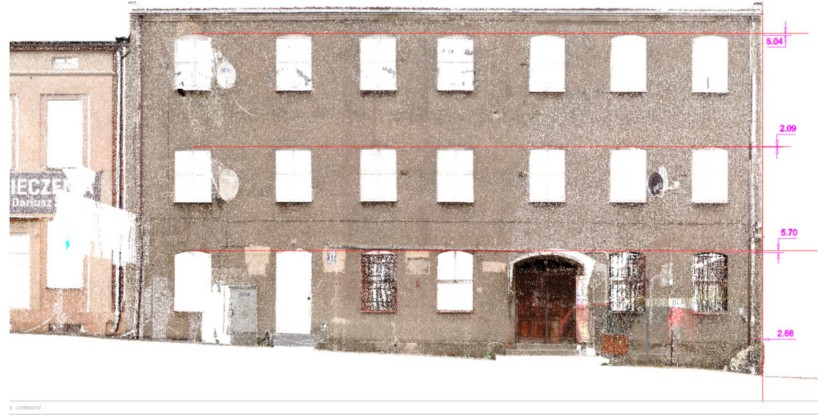

**Figure 10.** Outside view of the building made using TLS in Autocad 2015—Front wall with measured deformation [cm].

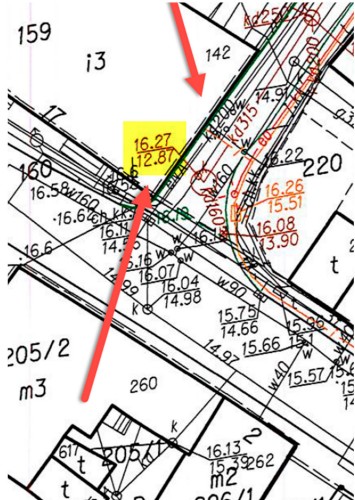

**Figure 11.** Ground work map—ordinate of replaced pipe near studied building.

Thanks to TLS technology and internal measurements (Figure 12), it was possible to prepare the cross-section of the building. The method allowed to perform observations of floor deflections and the overall geometry of the building. The measurements also show the displacement of the building towards Street (south-eastern direction). Additionally, large deflections of the floors were found. The first ceiling exhibit a deflection of 1.39 cm on the span of 400 cm. The calculated value of the deflection was already L/288, even without the full imposed load. Due to the load condition of the floor during the diagnostics (not fully loaded), L/350 was assumed as the reference value instead of L/250, which would result in a 1.14 cm limit value. In another part of the floor, the deflection of 1.80 cm on the length of 246 cm was measured, which results in L/132. The results show exceeding of the serviceability conditions. This conclusion was confirmed by measurements made in other places and sections of the building. The design of the floor does not meet the design requirements. The flaw occurred during the renovation works after the war. The calculations for existing floor beams shows that the safety requirements were not met.

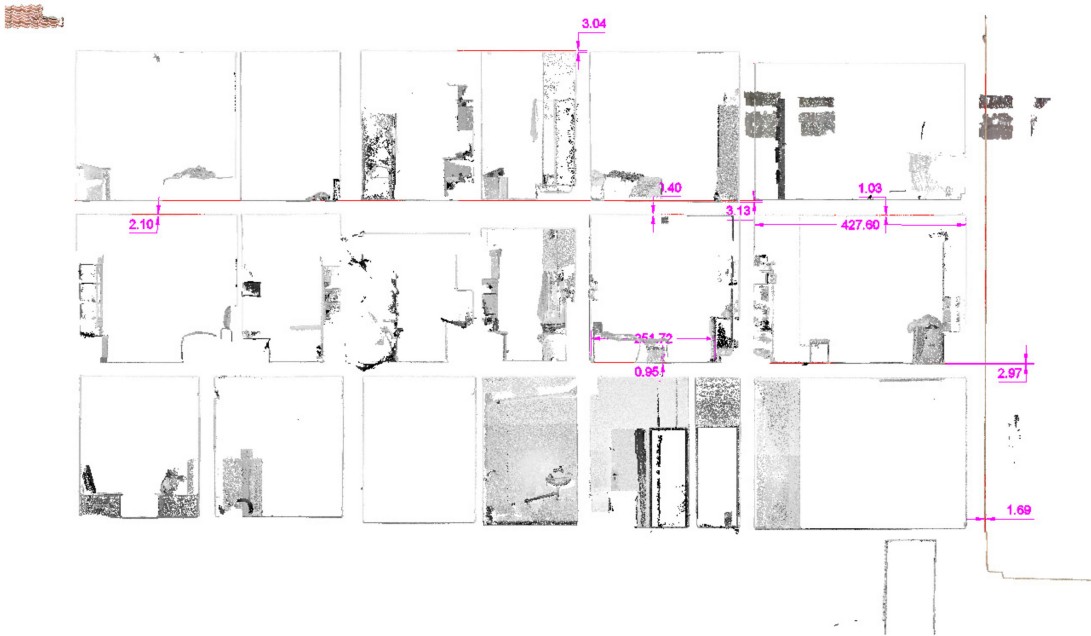

**Figure 12.** Internal view of the building made using TLS in Autocad 2015—cross-section with visible deformations (cm).

Another issue is the fire safety of the existing floors. A steel floor has practically no fire resistance and can quickly be destroyed in the event of a fire due to yielding. Steel loses its basic strength properties at the temperature of 600 °C, while a typical fire temperature is around 1200 °C. Originally, the tenement house certainly had wooden ceilings floor beams filled with the so-called improvement, which was a much better solution.

## 5. Proposed Repair Solutions and Conclusions

As a result of the analyses and diagnostics, it was proposed to suspend the investment (adjustment of the ground floor for the function of a kindergarten) in order to make further observations of the building. A single measurement does not provide a way to say whether the current state is stabilized or whether the building will exhibit further displacement. Further investigation into the building's movement might prove that errors were made during previous ground work. For such a result, the ground needs to be strengthened first.

It was proposed to reinforce the floors by welding IPE 120 profiles to existing IPE 140 steel beams at the bottom of the residential dwellings. The solution should provide the necessary strength to the structure. If possible, reverse deflection should be imposed on the beams to reduce the deflections in the future. A deflection of about 5–7 mm was assumed as sufficient. The welding of profiles can only be carried out if the entire floor slab on all floors has been stamped beforehand. This is necessary because the steel profiles may yield during welding due to high temperatures. Alternatively, steel channels 20 × 50 mm (50 mm of height) can be welded to the beams to increase the load-bearing capacity. The solution can be applied only if initial profiles are already covered with concrete (preventing loss of stability). This will increase the floor height by 5 cm. At the same time, the filling panel of the beam and block floor should be replaced with lighter material, e.g., mineral wool.

Steel channels should be supported through plates and not directly on the masonry. The solution prevents wall shearing in the future. Concrete cushions in these locations may also be considered.

The load-bearing walls near the location of cracks must be reinforced with additional steel bars $\phi$8 BSt500S L = 2 m, in every second joint on both sides. It is also possible to consider rebuilding of the cracked fragments.

Incorrectly made connection of building walls (vertical cracks visible from the outside) should be repaired by rebuilding the walls and adding reinforcement φ8 BSt500S L = 2 m in every second joint. The dimensions of bars should be determined with regard to existing wall fragment.

Steel beams in the building should be improved by increasing their fire resistance class to R60, e.g., by covering with fire-proof paints.

The foundations should be reinforced by adding a concrete tie beam around the building at the level of foundations or at the level of foundation walls that would protrude at least 30 cm from the existing geometry. The work should be carried out in a sectional manner, which would prevent the soil from displacing the foundations.

Due to the lack of tie beams that would stabilize the structure, a steel frame must be made from the outside. The frame should be fixed to the walls with chemical anchors and tensioned with bolts (Figures 13 and 14). Three sides are accessible from the outside while the fourth needs to be reached from inside of the building. Designed reinforcement of steel channels C120 was limited to a L80 × 60 × 7 angle bracket at the location of passing through the dwelling. Tensioning joints with 2 cm range were designed every 5 meters. These joints were also designed at each corner of the building.

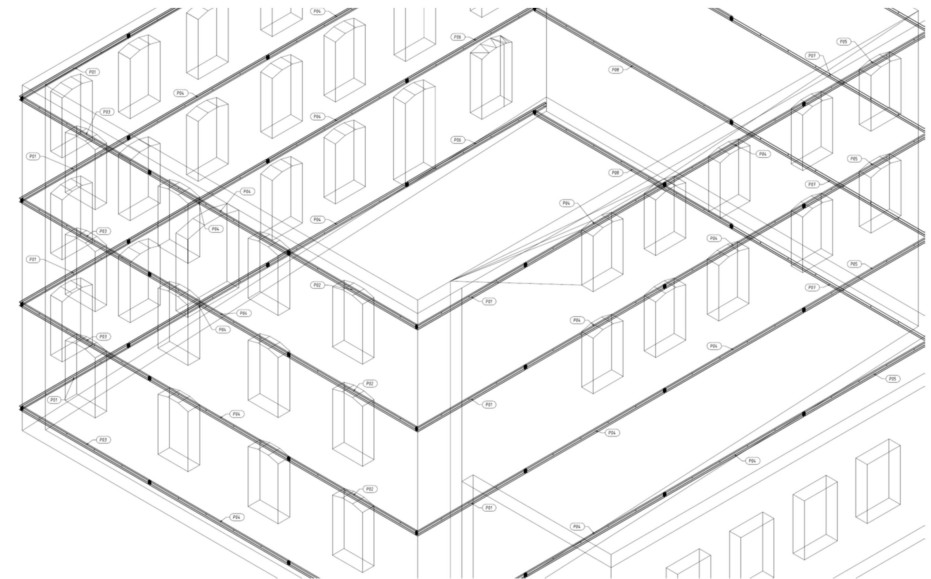

**Figure 13.** Proposed reinforcement solution for studied building—axonometry.

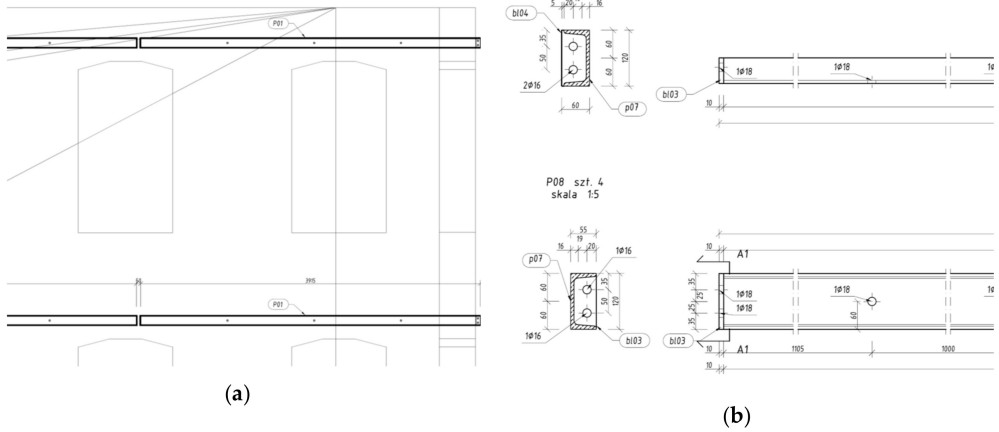

**Figure 14.** Proposed reinforcement solution for studied building: (**a**) front; (**b**) details.

The 3D model of the structure used in the design process was created in Autodesk Structural Detailing on the basis of the 3D model of the building from the RCP-type point cloud, previously used for building diagnostic measurements. Building walls were modelled as SOLID type elements.

## 6. Summary and Discussion

TLS measurements allowed for effective diagnostics and assessment of the technical condition of the building. In addition, they allowed to determine the causes of damage by understanding the behaviour of the entire structure.

It is important to identify the causes of damage while preparing for repair because this is the only way to design a proper and effective solution for the structure.

The article proves the usefulness of TLS measurements in diagnostics of structures. The obtained data in combination with BIM or CAD 3D programs significantly improved the quality of the work and the quality of the performed reinforcement projects.

The analysed building is in a poor technical condition and requires a number of repairs. The causes are the numerous mistakes made during the reconstruction of the building in the 1950s. In addition, recent land work in the immediate vicinity of the building caused its additional degradation—leaning and uneven settlement. It was not possible to check whether the condition is currently stable.

For similar building diagnostics, geometric data can also be obtained using total stations and polar measurements. However, the amount of data obtained will be much smaller. Determination of external and internal geometry of the building would be time-consuming and would be burdened with a greater measurement uncertainty. The use of TLS measurement tools provides a virtually complete building geometry and eliminates the possibility of overlooking important problems such as excessive floor deflection or uneven building subsidence.

An important aspect of the building analysis is its adjustment of the current fire protection requirements. The steel structure in the building requires fire protection to a minimum of R60 class.

**Author Contributions:** Conceptualization, R.N., R.O., R.R.; methodology, R.N., R.O., R.R.; validation, R.N. and R.O.; formal analysis, R.N., R.O., R.R.; resources, R.N. and R.O.; writing—original draft preparation, R.O.; writing—review and editing, R.N. and R.R. All authors have read and agree to the published version of the manuscript.

**Funding:** This research received no external funding.

**Conflicts of Interest:** The authors declare no conflict of interest.

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
