# Peer review of "Use of TLS (LiDAR) for Building Diagnostics with the Example of a Historic Building in Karlino"

_buildings, doi:10.3390/buildings10020024_

Round 1

Reviewer 1 Report

the topic is interesting and useful for historical heritage researchers. The method used is not an original method developed by the author. The method is generally used before the restoration works of historical buildings. The summary of the study can be developed for the purpose.

Author Response

Dear Sir/Madam,

thank You for reviewing our article. There were no questions that needed our response or comment. Agreed, LiDAR methods are very useful for examination of historical buildings.

Best regards,

Reviewer 2 Report

The scientific article presents the use of TLS data to diagnostic a historical object. In general, the topic is extremely interesting but also demanding and difficult. The scanning data allow capturing complete information about the object, however diagnostic requires data with high reliability and geometric correctness. Below I have listed comments and suggestions for Authors.
1. I suggest changing the title from LIDAR (Light Detection and Ranging) to TLS (Terrestrial Laser Scanning). LiDAR is usually used as aerial scanning.
2. The article lacks solid theoretical introduction. The state of the art should describe both the problem of buildings diagnostics (not only in the statement that it describes [1-14]) as well as the use of 3D scanning in object diagnostics. It should be written how such diagnostics are performed, to what extent, what is better in the described solutions and what is worse.
The introduction should usually be summarized by the purpose and thesis of the article. I suggest that the paragraph in lines 42-49 be divided into: a paragraph about TLS in the study of deformation of objects (including historic ones) and a paragraph about the purpose of the work.
3. Paragraph in lines 51-54 should be preceded by an introduction in the style: The object of research was ... .
4. The information in lines 71-77 indicates a crack in the building facade when two objects join together. Why such a crack can not be considered as a dilatation, since it connects two objects with two independent foundations. In such a situation dilatation is a normal phenomenon.
5. Line 155-156: Please give examples. Currently, most CAD programs import point clouds efficiently as a reference. What do the authors think about the free Cloud Compare program, which allows the implementation of publicly available PCL library algorithms?
6. Paragraph 169-173 reproduces the information already contained in the introduction, please move it to the introduction or simply delete.
7. The information contained in the paragraph from lines 199-201 should precede the description of the results from point cloud analyzes. Moreover, reference objects (white spheres, black and white targets) that were tie points during measurement and registration should be mentioned. What are the results of point cloud registration (RMS, MAE)?
Without analyzing the accuracy of the point cloud after registration, you cannot draw conclusions about the deformation of the object. Perhaps the deviations of the walls, building etc. are due to measurement errors with a poor opinion scanner, and not to the actual condition of the building? Such point cloud reliability testing after registration may even rely on the dimensioning of reference objects (spheres), scanned from several scanning stations, and not used in registration. If the model of such a sphere will be consistent, then we can be sure that the 3D data are of adequate accuracy and the clouds of points have properly combined.

Thank you.
Best regards.

Author Response

Dear Sir/Madam,

thank You for reviewing our article. Thank you for all of the suggestions.

As for discussion problems:

The authors agree with reviewers suggestion, the topic was changed to: “Use of TLS (LiDAR) for building diagnostics on the example of historic building in Karlino”. Corrected the article according to provided suggestions Corrected the article according to provided suggestions Agreed, but not in this particular case. The building’s structure was designed in a flawed way, without considering proper vertical dilatation of the construction. This could be performed by making doubled gable wall or concrete frame near dilatation. The structure was designed as one, without visible separated zones. The roof was supported on the whole structure, same case with slabs. In this particular case such dilatation would not provide structural stability. In my case I’ve mostly used Civil Engineering software such as Autocad, Revit, ArchiCAD, Allplan, Gstar Cad, Rhino. Only in Revit it was possible to load any size Point Cloud without any problem as the process is made dynamically (even 200 GB Point Cloud at only 32 GB of RAM). Allplan and GStar Cad cannot handle any of Point Cloud yet. ArchiCAD can handle only very small and light Point Clouds. To even operate you need to decrease the resolution. Rhino is quite good in this matter but also limited to RAM of computer as in does not load dynamically. As for Cloud Compare, the software appeared to have problems with RAM = limiting the size of Point Clouds. The only reasonable software I knew how to operate was Revit that handled this freely. I’ve read that Sketchup handles Point Clouds but only of limited size. That’s why we express the opinion that there is still a big problem with proper handling of Point Clouds. The standard computer limit of 32 GB of RAM is problematic for scans that exceed the size of 32 GB. Some of my building scans take 200 – 300 GB of space, it is hard to get such computer for a typical user. Normal computer can handle up to 32 GB of RAM (there is Motherboard limit for this). For any other purpose it is required to buy a special equipment like Work Stations. Didn’t find this inside introduction, maybe You meant the Abstract. For now we leave this in original position. Please check this again and let me know if this still needs correction. The problem described in our case study is a real design issue. Originally, an attempt was made to determine the geometry of the building using a total station. However, the results obtained did not allow to fully understand the state of the building’s structure. Further measurements using this device were considered. Attempts to put the measurement matrix inside the building in order to perform the measurements and combine them with the results of external dimensions measurements aroused great reservations. Initial analysis of errors of measurement works in the scope of transfer of errors in setting up subsequent measurement bases showed that the final accuracy of measurement works is practically of the same order as the expected deformations of the examined object. Such a state could not be accepted. The decision to use other measurement methods was natural. TLS measurements were practically the only alternative. The conducted TLS measurements and analysis of errors of these measurements showed a definite advantage of this method over total station measurements for the analyzed case. The specified errors of the TLS measurement process are based on the manufacturer's data and data received from the scanner and calculations performed by his software. Unfortunately, the manufacturer does not make available the calculation algorithms used to determine the measurement uncertainty in the assembly of individual scans into the final model. The authors plan to develop and execute a TLS test field for any model of scanners allowing to verify the final measurement errors obtained during the process of assembling scans into a larger model of building structure. In the measurement process, the function of combining scans offered by the manufacturer of measuring equipment was used. The scanner software, after the process of combining the scans, makes calculations and provides the errors of combining individual scans individually and in total for the whole model. The obtained error values did not exceed 3 mm, which is, in our opinion, a fully acceptable value in relation to the diagnosed values of structure deformations. The used scanning parameters for a given case are ½ or ¼ resolution of the reference scan depending on the distance from other scans. We have also introduced an appropriate addition to the article on this subject. Measurements of external walls were covered by single scans as well as ceiling deflections, hence the reading of their deformations is not tainted by the error of mutual assembly of scans from the scanner.

Best regards

Reviewer 3 Report

The paper is an interesting case study.
It is a good practice of a 3d laser scanning survey but its scientific contribution is very low because it does not add anything new to the state of the art.

We ask to explain the originality of the contribution and to integrate the data processing (point cloud processing, 3D modelling, the accuracy of the topographical network, tolerance in the measure, ....)

Author Response

Dear Sir/Madam,

thank You for reviewing our article.

The article discusses the practical application of TLS measurement methods for the analysis of historical buildings. Originally, the diagnosis of the building was attempted to be carried with a total station using the polar method. However, the obtained measurement results did not give a full picture of the current geometry of the whole object. Attempts to establish a measuring matrix inside the building to perform the measurements and combine them with the results of external dimensions measurements aroused great reservations. Initial analysis of errors of measurement works in the scope of transfer of errors in setting up subsequent measurement bases showed that the final accuracy of measurement works is practically of the same order as the predicted deformations of the examined object. Such a state could not be accepted. The decision to use other measurement methods was natural. TLS measurements were practically the only alternative. The conducted TLS measurements and analysis of errors of these measurements showed a definite advantage of this method over total station measurements for the analysed case. The specified errors of TLS measurement process are based on manufacturer's data and parameters obtained from the software of the scanner used. Unfortunately, the manufacturer does not make available the computational algorithms used to determine the measurement uncertainty when assembling individual scans into the final model. The authors plan to develop and execute a TLS test field for any model of scanners allowing to verify the final measurement errors obtained during the process of assembling the scans into a larger model of building structure. We agree that the methods of both total station and TLS measurements are known and applied, however, the procedure demonstrating the usefulness of applying TLS measurements to historic buildings whose geometry may be strongly unpredictable is an element of novelty of the presented case study. Moreover, the use of lintel lines and entire sections of existing buildings as reference lines in the analysis of building deformations is an approach unprecedented in previous case studies of this type. Such an approach allowed for a better understanding of the existing state of the building structure.

The article was extended – the reasons for choosing the TLS were added.

Best regards

Round 2

Reviewer 2 Report

Thank you very much for the detailed reply. There are no suggestions or comments for this article.
Performing analyzes of object deformation using TLS usually from the entire coherent point cloud (RMS registration accuracy below 1 mm), the cutter usually a fragment of e.g. the beam I am interested in, e.g. crack and PCL algorithms analyze changes in time. The fact that for the entire object TLS data with the correct resolution spoils the comparison process, but the fragments are not.
Best regards.